# Single-pixel infrared imaging thermometry maps human inner canthi temperature

Cheng Jiang[1,2], Patrick Kilcullen [1,2], Yingming Lai[1], Tsuneyuki Ozaki[1] & Jinyang Liang [1] ✉

Efficiently and accurately mapping the temperature of human inner canthi is crucial for disease diagnostics and monitoring. The specific anatomical location of the inner canthi precludes temperature screening methods that are invasive, require tissue contact, and/or demand active illumination. Camera-based thermography, although capable of passive and non-contact temperature mapping, suffers from low efficiency in pixel allocation to the inner canthi as well as from measurement inaccuracies due to background blending and moderate pixel sensitivity. In response to these challenges, we develop single-pixel infrared imaging thermometry (SPIRIT). We design diagonally aggregated two-dimensional transmissive encoding masks using a cyclic S-matrix, which supports compressed data acquisition in a single scan and high image quality through non-iterative reconstruction. SPIRIT maps the temperature distribution of human inner canthi with a resolution of 0.3 °C, which enables human temperature mapping via single-pixel imaging. Using SPIRIT, we reveal sub-degree temperature differences induced by daily physical activities and the glasses-wearing habit. These findings shed light on SPIRIT's contribution to improving evaluation criteria for public health, including COVID-19 febrile screening.

Efficient and accurate determination of core body temperature is of paramount importance for diagnostics and monitoring of many diseases, such as COVID-19[1,2], sepsis[3], neurological injuries[4], and hyperthermia[5]. Nonetheless, the measurements at the wrist, forehead, and armpit may diverge from core body temperature due to external influences (e.g., airflow and coverings) as well as variations in blood flow, heat dissipation, and insulation[6]. By contrast, the inner canthi of the human eyes, shielded within the eye sockets, are situated in a more stable environment[7]. Meanwhile, they have efficient and rapid heat exchange with blood via a prominent vascular network, which enables a sensitive response to core body temperature[8]. The thin, transparent mucous membranes of the inner canthi also facilitate an accurate temperature reading[9]. To determine the inner canthi temperature, international standards recommend spatially resolved measurements with at least $3 \times 3$ pixels, each being $1 \times 1 \, mm^2$ in size[10–12].

Existing techniques confront challenges in accurately mapping the inner canthi temperature. The requirement for fast and efficient screening excludes all invasive methods that rely on target labeling (e.g., using rare-earth doped nanoparticles[13,14] or thermochromic liquid crystals[15]) or target coupling (e.g., using ultrasound gel[16]). Moreover, for eye safety concerns, techniques necessitating active illumination, including laser thermometers[17], photoacoustic imagers[18,19], and Raman spectrometers[20,21], are also inapplicable. Furthermore, to avoid clinical complications, techniques requiring direct tissue contact, such as thermocouples[22] and thermistors[23], are disqualified.

Passive, non-contact techniques represent the most promising approach to mapping temperature distribution in the inner canthi. Among them, imaging thermography using a two-dimensional (2D) CCD or CMOS sensor, which can offer temperature information over a large spatial format, has become mainstream[24,25]. Narrow-gap semiconductors, such as mercury cadmium telluride (HgCdTe), are

[1]Centre Énergie Matériaux Télécommunications, Institut National de la Recherche Scientifique, Université du Québec, Varennes, QC, Canada. [2]These authors contributed equally: Cheng Jiang, Patrick Kilcullen. ✉e-mail: jinyang.liang@inrs.ca

particularly sensitive to long-wave infrared (LWIR) radiation[26] and hence are used as the photosensitive component in each sensor pixel to detect the thermal radiation emitted from humans[27]. Though safe and easy to implement, this technique faces challenges in practicality and measurement accuracy. In particular, to minimize thermal noise, 2D LWIR sensors require cooling systems with efficient heat dissipation[28,29]. However, commonly used technologies, including Stirling cryo-coolers and liquid coolers, introduce complexities in operation and considerably increase the cost[30]. Moreover, vibration induced by the compressor assembly adversely affects the imaging quality and sensor durability[31]. In practice, LWIR thermography is typically configured to image the entire human face. The small size and spatial separation of the inner canthi result in a considerably low portion of allocated pixels (e.g., <0.5%) for temperature mapping[12,32]. Consequently, the measurements are unavoidably affected by the background blending with the targeted areas. Considering the temperature-signal correlation, any perturbation in the signal inevitably compromises the accuracy of temperature readings[33].

Single-pixel imaging (SPI) offers a promising alternative to overcome the limitations of imaging thermography. This technique encodes the object's radiation by using pre-determined spatial patterns, with subsequent data acquisition performed by a point detector. During the ensuing image reconstruction, these encoding patterns serve as prior knowledge, enabling the accurate retrieval of spatial information[34,35]. The throughput and multiplexing advantages embedded in SPI improve signal-to-noise ratios (SNRs) in measurements and reconstructed images, enabling it to detect fine intensity gradients and spatial details[36].

SPI leverages the advantages of single-pixel detectors with their wider availability in the sensing spectrum to fill the void where the 2D-sensor-based counterparts are not available or impractical[37], including infrared[38,39], terahertz[40], and acoustic[41] bands. Moreover, compared to the complex heat-management systems required for 2D sensors, high-quality cooling of a single-pixel detector is more feasible, reliable, and economical, which enhances the accessibility and maintainability of the SPI systems. Furthermore, the masks used in SPI can be designed to image customized fields of view (FOVs). This resource-conserving paradigm allows for focusing on the regions of interest, which thus avoids capturing and processing irrelevant information.

Despite these attractive advantages, to date, SPI has not yet been demonstrated for human thermometry. A pivotal challenge lies in the ineffectiveness of available spatial light modulators (SLMs) in modulating LWIR light[42–45]. These pixelated devices, whose pixel size is comparable to the wavelength of LWIR light, bring in considerable loss in diffraction[46]. Their protective windows are opaque to LWIR light[47]. Efforts of re-windowing with zinc selenide (ZnSe) have enabled LWIR modulation using a digital micromirror device[48]. However, window replacement can induce physical stress, contamination, and thermal mismatch, which can lead to misalignment, warping, or cracking of the delicate micromirrors[49]. As an alternative, SLMs are capable of delivering patterned visible or near-infrared pulses to a semiconductor to induce transient absorption for LWIR light modulation[50]. However, the required high-energy pump sources[51] largely reduce its application scope. Besides the inherent limitations in SLMs, the coding strategy in most existing systems, which is based on the Walsh–Hadamard matrix[52], exhibits drawbacks from time-consuming, compressed-sensing-based image reconstruction using a large number of iterations[53]. Thus far, instant temperature mapping is still beyond reach.

To surmount these limitations, we develop single-pixel infrared imaging thermometry (SPIRIT) for temperature mapping in humans. This system passively detects thermal radiation using a transmissive encoding scheme. Diagonally aggregated physical masks constructed from a cyclic S-matrix are designed for 2D spatial encoding via a single compressed-sensing-compatible scan, ensuring that 100% of the

imaging pixels are dedicated to the targeted regions in the system's FOV. The encoded thermal radiation in the LWIR spectral range is then integrated by a cooled HgCdTe single-pixel detector. Leveraging spatial multiplexing, thermoelectric cooling, lock-in amplification, and non-iterative image processing, SPIRIT instantly maps the temperature distribution with a resolution of 0.3 °C. We apply SPIRIT to human inner canthi temperature mapping and monitor diurnal temperature variations. SPIRIT is also used to investigate the influence of gender and glasses wearing on the temperature readings at the inner canthi.

## Results
### System setup
A schematic of the SPIRIT system is depicted in Fig. 1. Thermal radiation emitted from the inner canthi is first imaged by a ZnSe lens (39-532, Edmund Optics) onto the intermediate image plane with a 4:1 magnification ratio. A mask plate and a window plate, each with a 2" diameter and a 150-μm thickness, are placed at the intermediate image plane. The mask plate, attached to a motorized translation stage (MTS25-Z8, Thorlabs) moving vertically, contains two encoded stripes separated by 7.5 mm. Each encoded stripe comprises aggregated masks with encoding pixels of 250 μm × 250 μm in size. The window plate has two open areas, measuring 2.75 mm × 1.75 mm and 2.75 mm × 1.50 mm, rotated to −45° and 45°, respectively, and separated by the same distance as the two encoded stripes (i.e., 7.5 mm). In this way, thermal radiation is encoded by two masks with sizes of 11 × 7 and 11 × 6 pixels, respectively (see the blue box in Fig. 1). Subsequently, the encoded thermal radiation in both areas is collected by another ZnSe lens (307B/1.0, Yoseen Infrared), passing through an optical chopper (MC2000B, Thorlabs), and finally focused onto an HgCdTe photodiode (PDAVJ10, Thorlabs) cooled by a Peltier thermoelectric cooler. SPIRIT's sensing spectrum, limited by the anti-reflection coating of the second ZnSe lens and the responsive range of the photodiode, is from 8 μm to 10.6 μm. As depicted in the yellow dashed box in Fig. 1, the FOV in this configuration contains two regions—11 mm × 7 mm (−45°-rotated) and 11 mm × 6 mm (+45°-rotated) in size—separated by 30 mm, which can cover both inner canthi of most humans[54]. Meanwhile, each encoding pixel samples a 1 mm × 1 mm area in the targeted inner canthus, which satisfies the recommendation specified in the international standards[10,11]. Details of the mask-window alignment are presented in Supplementary Note 1 and Supplementary Fig. 1.

In operation, the optical chopper provides an 850-Hz modulation. The acquired signals are sent to a lock-in amplifier (SR830, Stanford Research Systems) to enhance the SNR before being stored in a digitizer (ATS9625, Alazartech). Data acquisition is synchronized with the vertical scan of encoding masks. The resulting data points, termed "bucket signals", are then transferred to a computer for image reconstruction. Each scan, collecting 91 bucket signals (detailed in the next section), takes ~15 s. A visible color camera (16-544-RCD-05P, Edmund Optics) with a camera lens (MVL8M23, Thorlabs) is also used to coincide the inner canthi with the FOV of the SPIRIT system. Details of system synchronization and FOV co-registration are provided in Supplementary Notes 2–3 and Supplementary Figs. 2–3. A performance comparison highlighting the impact of the lock-in amplifier on SNR enhancement is presented in Supplementary Note 4 and Supplementary Fig. 4.

### Coding strategy
SPIRIT's encoding patterns are meticulously designed to efficiently measure temperature distribution from both inner canthi with a balance between data acquisition time and reconstructed image quality. The bucket signals, represented by an $m$-element vector $y$, can be interpreted as the inner products computed optically between an image comprising $n$ pixels, represented by $x$, and the collection of 2D encoding patterns. This procedure can be succinctly articulated as

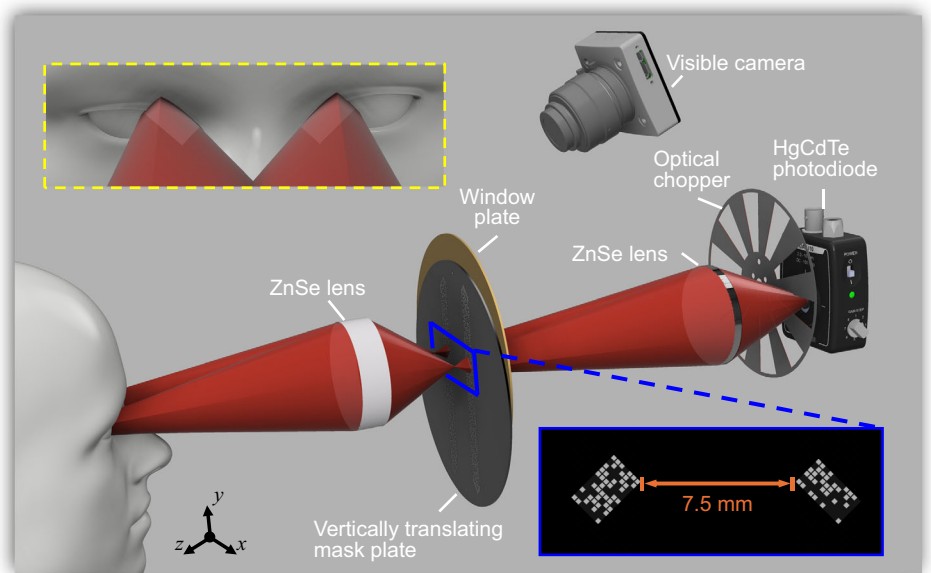

**Fig. 1 | Schematic of single-pixel infrared imaging thermometry (SPIRIT).** Yellow dashed box: Close-up view of the field of view. Blue box: Close-up view of an encoding mask passing the open areas on the window plate. The mask plate and the window plate, although made from the same material, are depicted in different colors for display purposes. The CAD sketches of the optical chopper, photodiode, and visible camera lens in this figure are courtesy of Thorlabs, Inc. The 3D head model is reproduced with permission under a CC BY 4.0 license from Sketchfab.

$y = Sx$, where the measurement matrix $S$ incorporates each encoding pattern represented in row form.

SPIRIT's encoding patterns are derived from a cyclic S-matrix (detailed in Methods, Supplementary Notes 5–6, Supplementary Fig. 5, Supplementary Table 1, Supplementary Movie 1, and Supplementary Software Package), which is known to offer superior performance over random-matrix encoding in terms of noise reduction and computation efficiency[55]. As depicted in Fig. 2a, SPIRIT's measurement matrix $S$ operationalizes a binary cyclic S-matrix with twin-prime factors of $p = 11$, $q = 13$, yielding $n = p \times q = 143$. As an example, the first 2D encoding pattern, produced by row-major reshaping the first row of $S$, is shown in the blue shaded area of Fig. 2b. Then, several replicas of this pattern are tiled with shifts (shown in red, yellow, and green colors in Fig. 2b). This construction encapsulates the 2D encoding patterns reshaped from any row of $S$.

We develop a diagonal scanning approach for a compact aggregation of the encoding patterns compatible with compressed sensing in data acquisition and supporting non-iterative image reconstruction. As the first step, the tiled pattern is further duplicated and shifted (Supplementary Fig. 5d). Starting from the top-left corner, an aggregate pattern is generated by cropping out rectangles with $p \times q$ in size (delineated by the red dashed box in Supplementary Fig. 5d) along the diagonal direction across this matrix. As illustrated by the five colored boxes in Fig. 2b, this aggregate pattern, although corresponding to disconnected rows in the measurement matrix, supports continuous scanning (Supplementary Movie 1). More importantly, this approach supports a highly uniform sampling distribution of the set of 2D-shaped bucket signals. Figure 2c depicts the maximum distance of each unsampled signal to its sampled neighbors for $m = 91$ patterns. Compared to the conventional linear scan counterpart[37], SPIRIT substantially narrows the gaps between the unsampled signals. Considering the similarity between adjacent encoding patterns and thus their corresponding bucket signals, the unsampled data are estimated by using discrete Laplace interpolation[56], which enables forming an $n$-element vector $\tilde{y}$. Finally, by direct matrix inversion, the image is retrieved as $\tilde{x} = S^{-1}\tilde{y}$. This non-iterative compressed-sensing image reconstruction enables SPIRIT's online processing for instant data retrieval.

To implement the designed aggregate patterns for inner canthi, each encoding pattern is divided into two parts with the sizes of $p \times [(q+1)/2]$ and $p \times [(q-1)/2]$, respectively. Subsequently, each part undergoes flipping and/or rotation (detailed in Supplementary Note 6, Supplementary Fig. 5f, and Supplementary Movie 1), which better accommodates variations in the intercanthal distance among different individuals (Fig. 2d). By following the same arrangement procedure for each encoding pattern and exploiting the aggregation between consecutive encoding patterns, two encoded stripes are formed (Fig. 2e). A simulation of this coding strategy is presented in Supplementary Note 7 and Supplementary Fig. 6. Besides the $m = 91$ encoding patterns, four auxiliary L-shaped run-way patterns are appended at the beginning and the end of both encoded stripes to indicate the onset and completion of encoding as well as to facilitate the mask-window alignment. Finally, to ensure vertical alignment in encoding-mask scanning, one pixel is placed at the same horizontal position on the top and bottom of the mask plate (Fig. 2e).

## Quantification of the SPIRIT system's performance

To experimentally demonstrate LWIR imaging using SPIRIT, we captured thermal radiation penetrating through a visible-light blocker. As depicted in Fig. 3a, temporally stable and spatially uniform thermal radiation was emitted from a blackbody radiator (YSHT-35, Yoseen Infrared) with a temperature of 37.0 °C. Within SPIRIT's FOV, the thermal radiation was shaped by two metal plates marked with the hollow letters "X" and "Z", respectively. Moreover, a piece of black plastic cover, opaque to visible light but partially transparent to LWIR light, was placed in front of the SPIRIT system. A pre-calibrated thermal camera (M384D, Yoseen Infrared) was implemented to provide the ground truth of the thermal signals. The experimental results are shown in Fig. 3b. While the visible camera in the SPIRIT system captured no information due to the cover's opacity in the visible spectrum, the LWIR imaging provided by SPIRIT correctly discerned both letters.

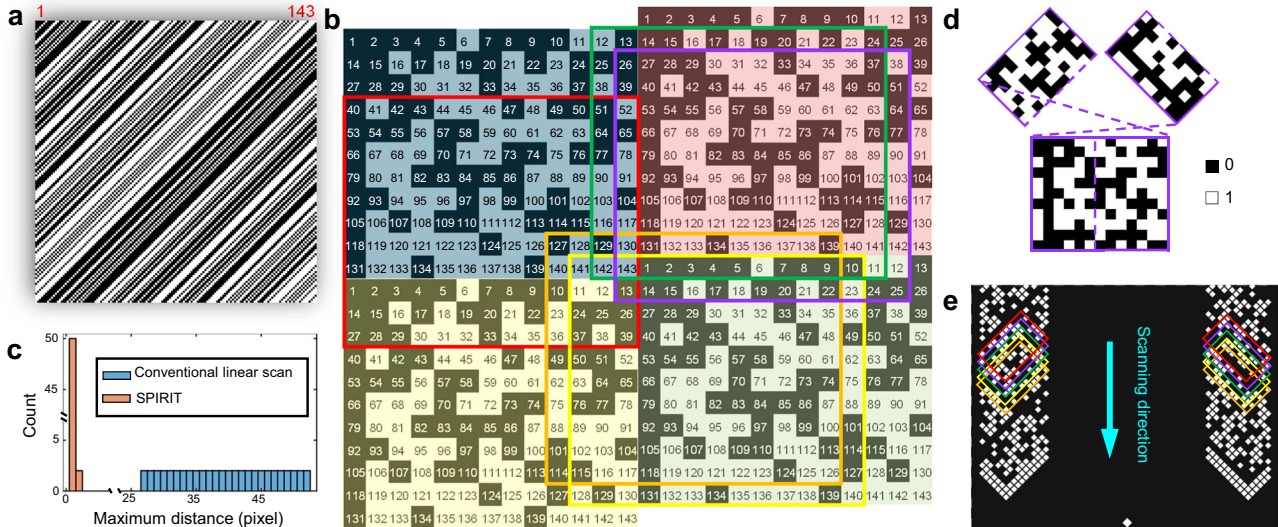

**Fig. 2 | Coding strategy of SPIRIT. a** Binary cyclic S-matrix of order $n = 143$ (i.e., $p = 11$ and $q = 13$). **b** Tiled matrix derived from (**a**). The numbers represent the corresponding row indices of each element of the first column of the matrix shown in (**a**), with shaded colors highlighting its tiling property. The colored boxes outline five consecutive encoding patterns corresponding to isolated rows in (**a**) indexed by their top-leftmost corner elements. **c** Histogram of the maximum neighboring distance of the conventional linear scan and SPIRIT with the same scanning steps. **d** Separation and rearrangement of a representative encoding pattern marked by the purple box in (**b**). **e** Beginning section of the mask plate. The colored boxes outline the masks generated by the corresponding encoding patterns marked in (**b**).

To investigate the spatial resolution of SPIRIT, we imaged a constant thermal signal from the blackbody radiator through a 0.5-mm slit rotated to four angles− 0°, 90°, 45°, and −45° (Fig. 3c–f). Line profiles of normalized intensity along the direction of the slit width were then extracted for each rotation position. Spatial resolution was determined by counting the number of pixels along the line profile whose normalized intensity exceeded 0.5. This evaluation reveals that SPIRIT's spatial resolution is determined by the finite size of the encoding pixels. All four orientations exhibit the same spatial resolution of one SPIRIT pixel despite the anisotropy caused by the pixel shape. Further details on the number of data points are provided in Supplementary Table 2.

After validating the imaging capacity of SPIRIT, we performed temperature calibration using the blackbody radiator whose spatially uniform temperature was tuned from 31.7 °C to 39.8 °C with a step of 0.9 °C. For each preset temperature on the blackbody radiator, five measurements were conducted. The intensity in the reconstructed image is correlated with the temperature by[57]

$$\widetilde{x} = C\varepsilon\sigma T_{s}^{4} + \Delta . \tag{1}$$

Here, $T_s$ is the preset temperature of the blackbody radiator, $\varepsilon = 0.98$ is the emissivity of the blackbody radiation source and of human[58]. $\sigma = 5.67 \times 10^{-8}$ W m$^{-2}$ K$^{-4}$ denotes the Stefan−Boltzmann constant. $C$ takes into account the emission area and light-to-voltage conversion, and $\Delta$ is the measurement error, both of which were determined by a pixel-by-pixel linear fitting using Eq. (1). The measured temperature map, $T_m$, was calculated as

$$T_{m} = \sqrt[4]{\frac{\widetilde{x}}{C\varepsilon\sigma}} . \tag{2}$$

The result is shown in Fig. 3g, which shows SPIRIT's temperature resolution of 0.3 °C.

Upon completion of temperature calibration, we used SPIRIT to reconstruct structured features and temperature distributions using Eq. (2). The blackbody radiator's temperature was set to 32.0 °C, 35.0 °C, and 38.0 °C, respectively. For each temperature, we placed two metal plates with engraved letters in the FOV. The resulting reconstruction images, presented in Fig. 3h, i, display excellent agreement with the ground truth, demonstrating SPIRIT's ability to robustly map the temperature of fine features.

**Temperature screening using SPIRIT**

To demonstrate SPIRIT of the human inner canthi, we recruited 39 volunteers from diverse ethnic backgrounds (detailed information is included in Supplementary Table 3). The thermal camera was deployed as the gold standard to monitor the facial temperature of the volunteers during SPI. Figure 4a shows the representative temperature screening results for four volunteers. SPIRIT's results overlay the eye contour of the volunteers generated from the images captured by the co-registered visible camera. Using images captured by the visible camera, the inner canthi are identified as the areas formed by the inner (or nasal) junction of the upper and lower eyelids[59]. SPIRIT's reconstructions reveal that the inner canthi regions exhibit higher temperatures relative to the adjacent areas covered by skin. The ground truth results are provided in Supplementary Note 8 and Supplementary Fig. 7, which show a good agreement with SPIRIT.

The mean temperatures within the left and right inner canthi were calculated by using only the pixels residing in these areas in the images reconstructed by SPIRIT (see the number of data points in Supplementary Table 4). Figure 4b shows that the 39 volunteers exhibit mean inner canthi temperatures ranging from 34.7 °C to 37.2 °C, aligning well with the ground truth. It is noteworthy that the mean temperatures of the two inner canthi can differ for the same individual, possibly due to asymmetrical blood flow, physiological stress, or emotional state[33]. Moreover, considering the recommended threshold temperature for fever screening based on inner canthi to be 37.1 °C[8], one volunteer (V16) was classified as fever-positive while the remaining 38 volunteers were fever-negative, all of which were consistent with the gold-standard measurements.

**Monitoring of temperature variation related to physiology and habits using SPIRIT**

We further investigated the temperature variation of the inner canthi related by three key factors among participants: diurnality, gender,

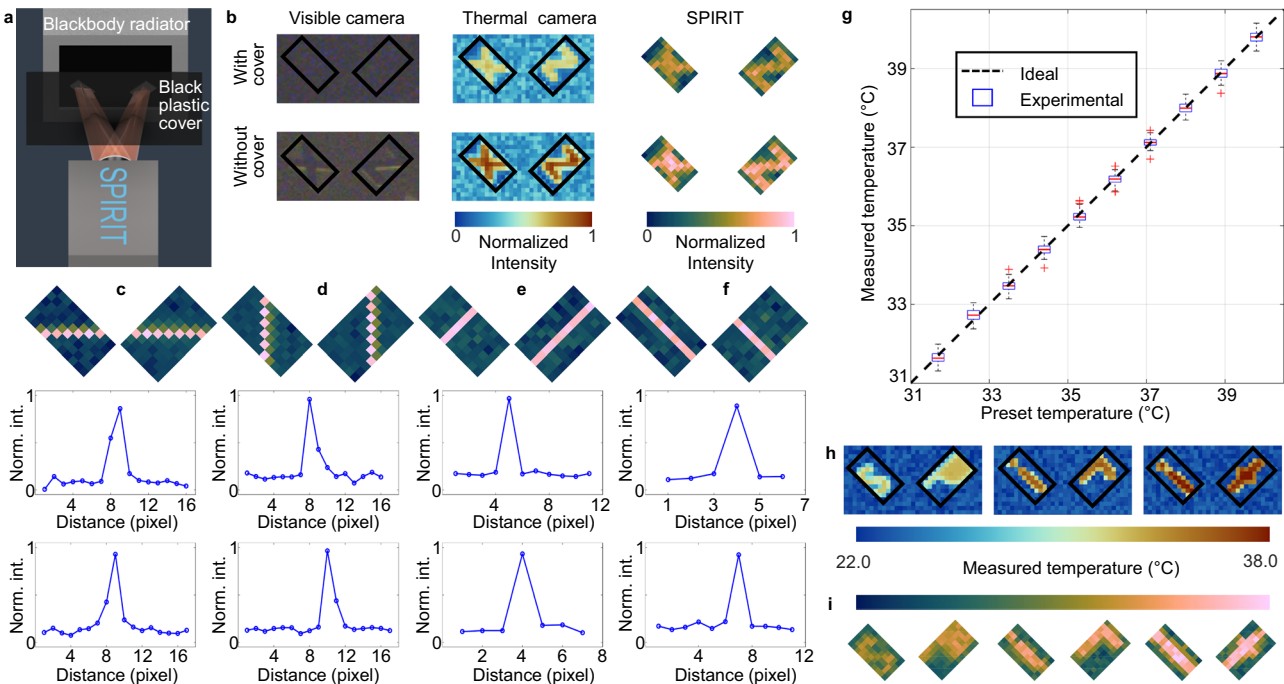

**Fig. 3 | Quantification of the SPIRIT system's performance. a** Experimental setup of the proof-of-concept demonstration. **b** Comparison of measurements using a visible camera (left), a thermal camera (middle), and SPIRIT (right), with (top row) and without (bottom row) a black plastic cover. Black boxes indicate the SPIRIT's field of view. SPIRIT reconstructions of a slit rotated to 0° (**c**), 90° (**d**), 45° (**e**), and −45° (**f**). Line profiles in the middle and bottom rows show the normalized mean intensity along the lines parallel to the slit in the left region and the right region of the field of view. The number of data points ($n_s$) in each line is provided in Supplementary Table 2. **g** Relationship between the measured and the preset

temperatures. For each temperature, $n_s = 715$. In the box plot, the center line represents the median, the top and bottom of the boxes represent the 75th and 25th percentiles, respectively, and the whiskers indicate the minimum and maximum values within the interquartile range. The red crosses refer to outliers, which are values more than 1.5 times the interquartile range away from the bottom or top of the box. **h, i** Ground truth of structured features with different temperatures (top row) and the corresponding reconstructions of SPIRIT (bottom row). Norm. int.: Normalized intensity.

and glasses wearing[8,60,61]. For volunteers who regularly wear glasses, measurements were conducted immediately after the glasses were removed. To examine the impact of diurnal changes on temperature, a total of 29 volunteers (the remaining 10 volunteers provided only one measurement due to personal availability) were measured across four time windows (i.e., 9:00–11:00, 11:00–13:00, 13:00–15:00, and 15:00–17:00), as shown in Fig. 5a. Measurements from the fever-positive individual (i.e., V16) were excluded from the ensuing analyses. The 28 fever-negative participants had an equal representation in gender (i.e., 14 males and 14 females) and for each gender, seven wore glasses and seven did not.

Diurnal temperature variations were examined using SPIRIT measurements taken from individuals over a single day. As an example, Fig. 5b shows the imaging results of volunteer V13. The measured temperature maps across these time points reveal an undulating temperature increase of 0.9 °C with a fluctuation cycle of 4 h, characterized by a rise of 0.7 °C in the morning due to increased physical activity and hence a higher metabolic rate, a decline of 0.4 °C around noon as a result of energy dips associated with digestion and midday rest, and a subsequent rise of 0.6 °C in the afternoon by an increased level of activities. This trend, shared in the measured temperature variation in all 28 volunteers (Fig. 5a), reflects human physiology in response to the body's varying needs and activities[62].

We further analyzed temperature differences related to gender and glasses wearing across each measurement time window by dividing the participants into four groups. For volunteers who do not wear glasses, the measured temperatures for females were 35.5 ± 0.1 °C, 36.0 ± 0.1 °C, 35.6 ± 0.1 °C, and 36.0 ± 0.1 °C (mean ± standard deviation) across the four measurement time windows, while the corresponding temperatures for males were 35.3 ± 0.1 °C, 35.8 ± 0.1 °C,

35.4 ± 0.1 °C, and 35.9 ± 0.1 °C, respectively. Similarly, for volunteers who wear glasses daily, the measured temperatures for females were 35.3 ± 0.1 °C, 35.7 ± 0.1 °C, 35.3 ± 0.1 °C, and 35.8 ± 0.2 °C, compared to 35.1 ± 0.1 °C, 35.6 ± 0.1 °C, 35.2 ± 0.2 °C, and 35.6 ± 0.1 °C for males. As shown in Fig. 5c, under the same glasses-wearing condition, comparisons reveal a consistent mean temperature difference of 0.1–0.2 °C between male and female volunteers across the four time windows, which is likely attributed to metabolic rates and hormonal factors[63].

As for the glasses-wearing-related temperature differences, two examples are shown in Fig. 5d, and the gender-specific statistical analysis is presented in Fig. 5e. Both comparisons reveal that volunteers who do not wear glasses exhibit a 0.2–0.3 °C higher temperature than glasses-wearing volunteers. The observed discrepancy in inner canthi temperatures could be attributed to several factors. First, a frame of glasses can exert slight, continuous pressure on the skin and underlying tissues at the bridge of the nose and around the temples[64]. Over time, this pressure could lead to minor reductions in blood flow to the area, slightly lowering the temperature of nearby regions including the inner canthi. Moreover, glasses can trap humidity in the ocular area, and as this moisture evaporates, it could cool the surrounding skin, affecting the inner canthi temperature[65]. Finally, wearing glasses changes how people perceive and react to their environment, potentially altering typical eye movements[66]. Reduced movement and the resulting decrease in muscular activity around the eyes might also have a minor effect on local temperature.

We performed statistical analyses of these measured results, shown in Fig. 5c and e. We first verified data normality with the Kolmogorov-Smirnov test. Next, a two-sample $t$-test was applied to establish significant differences ($p < 0.05$) between groups, followed by a right-tailed $t$-test to confirm that one group consistently exhibited

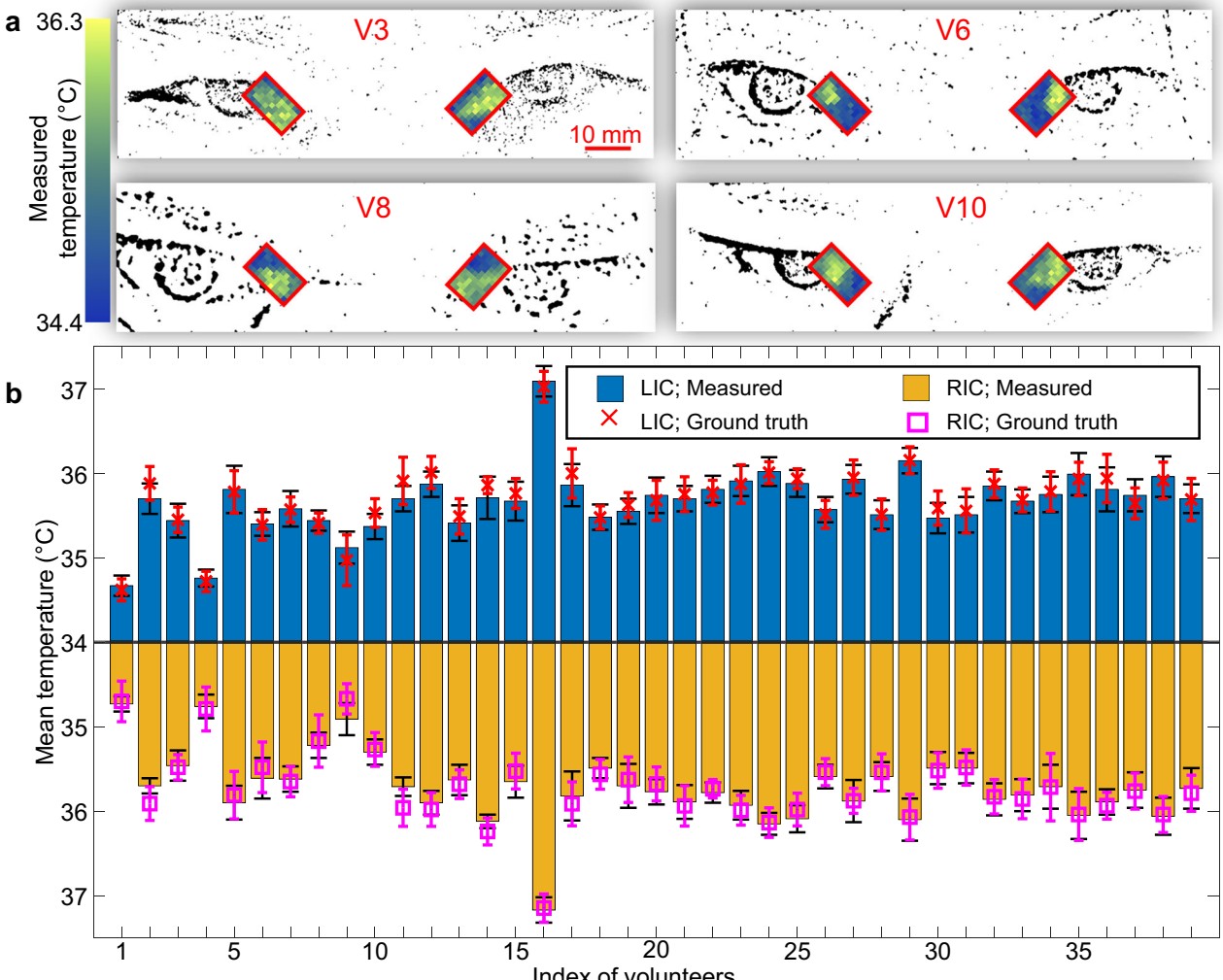

**Fig. 4 | Temperature mapping of human inner canthi by SPIRIT.**
**a** Representative reconstruction results for four volunteers overlaying the corresponding eye contours. **b** Comparison of the mean temperature of pixels within the left inner canthus (LIC) and right inner canthus (RIC) of 39 volunteers, measured by SPIRIT (shown as the bar heights) and the thermal camera as the ground truth (marked by the red crosses and the magenta squares). Error bars: standard deviation. The values of $n_s$ vary for each volunteer and are provided in Supplementary Table 4.

higher temperatures. These analyses confirmed that the observed temperature differences related to diurnality, gender, and glasses wearing are statistically significant. Details of these statistical analyses are included in Supplementary Note 9, Supplementary Figs. 8–10, and Supplementary Tables 5–6.

## Discussion

We have developed SPIRIT for efficient and accurate temperature mapping of human inner canthi. SPIRIT incorporates diagonal aggregation of 2D encoding patterns to uniformly fill the 2D-reshaped bucket signals, which allows non-iterative image reconstruction for instant data retrieval. SPIRIT has enabled passive temperature mapping of human inner canthi in 15 s with an overall frame size of 11 × 13 pixels and a temperature resolution of 0.3 °C. We have also applied SPIRIT to the monitoring of the diurnal temperature variations as well as an investigation of temperature differences related to gender and glasses wearing.

SPIRIT possesses several technical advantages. First, its transmissive masks with a relatively large encoding pixel size reduce diffraction loss and the number of required optical components, which enhances light throughput and hence the system's sensitivity compared to SLM-based SPI. SPIRIT also employs a single compressed

sensing-compatible scan of diagonally aggregated 2D masks generated from a cyclic S-matrix. This coding strategy not only accommodates variations in inner canthi structures and intercanthal distances but also facilitates lightweight, non-iterative image reconstruction. Moreover, in contrast to camera-based 2D thermography techniques, SPIRIT dedicates its entire FOV to the inner canthi, which enhances the measurement efficiency and reduces inaccuracies in temperature readings due to the background blending.

SPIRIT also holds practical merits. Its passive and non-contact operation is eye-safe and avoids sanitizing complications, making it suitable for large-scale temperature screening. Moreover, considering the international regulation and control of 2D thermal sensors, SPIRIT offers an open-source solution[67]. Its ability to detect sub-degree temperature differences related to daily physical activities, gender, and the glasses-wearing habit sheds light on setting more precise evaluation criteria for public health (e.g., for COVID-19 febrile screening). Notably, SPIRIT unexpectedly revealed the onset and development of fever in one volunteer (V16), who started with the fever-negative condition (Fig. 5a). As another example, the glasses-wearing-related temperature difference provides experimental evidence for the under-explored impact of glasses-established microclimate between the inner canthi and the spectacle lenses[68]. Its economic setup and instant data

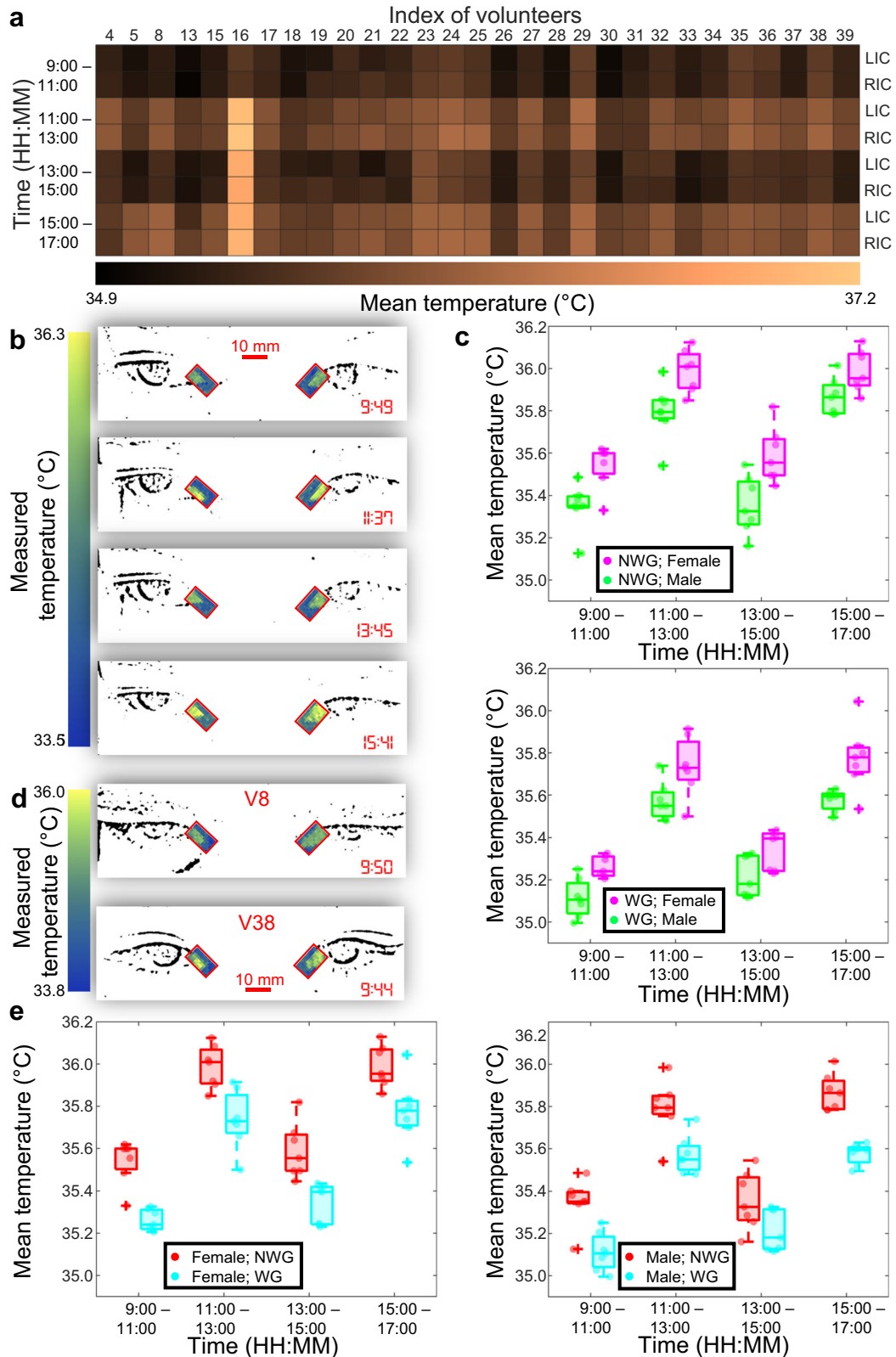

**Fig. 5 | SPIRIT of human temperature variation. a** Heatmap of inner-canthus temperature across four measurements for 29 volunteers on a single day. **b** Temperature maps of volunteer V13 at four time points on a single day. **c** Gender-related temperature differences among volunteers, subdivided into those not wearing glasses (NWG, top) and those wearing glasses (WG, bottom). For both plots in (**c**), $n_s = 7$ for both female and male. **d** Representative results of volunteers with (top) and without (bottom) glasses. **e** Glasses-wearing-related temperature differences among volunteers, subdivided into females (left) and males (right). For both plots in (**e**), $n_s = 7$ for both WG and NWG. The box plots of (**c**) and (**e**) follow the same convention as that used in Fig. 3g.

processing further strengthen its implementation perspective in public places and clinics.

As a generic platform, SPIRIT holds promises for future development and applications. Currently, the SPIRIT system possesses two technical challenges. First, compared to infrared thermography, SPIRIT requires precise alignment of various components for data acquisition and relies on image reconstruction to retrieve temperature maps. Second, the 15-s data acquisition time requires volunteers to keep their eyes open for the entire duration, which sometimes caused eye blinking, leading to unsuccessful data acquisition. To overcome these limitations, several technical improvements are envisioned. First, an automated calibration module using fiducial markers and real-time motorized adjustments for mask-window alignment could simplify SPIRIT's component alignment[69,70]. Second, a fully automated inner-canthus detection algorithm could be developed to enhance the temperature measurement process[71–75]. Third, a high-speed rotating mask could be employed to increase frame size and reduce data acquisition time[76–78]. Fourth, advanced interpolation methods could be explored to potentially improve reconstructed image quality without compromising processing time[79,80]. Finally, a zoom ZnSe lens[81] could be engineered to adjust the magnification ratio so that the intercanthal area projected onto the mask plate would maintain a constant length (i.e., 7.5 mm), enhancing SPIRIT's adaptability to a wide range of facial geometries. Besides temperature screening, SPIRIT is readily applicable to perceiving abnormal temperature signatures from diseases (e.g., inflammation[82], circulatory disorders[83], and breast cancer[84]) and detecting objects in obscured conditions (e.g., fog[85], smoke[86], and occlusion[87]). Other potential applications also include infrared astronomy, atmospheric observations[88], and property characterization of advanced materials[89].

# Methods

## Human study

The human study was conducted in strict adherence to the protocol (Project CÉR-20-578) approved by the Human Ethics Research Committee at the Institut National de la Recherche Scientifique, Université du Québec. In total, 39 volunteers were recruited for this study, including 19 females and 20 males, as determined by self-reported gender. Informed consent was obtained from every participant. Additional information in each experiment presented in this work is illustrated in Figs. 4–5 and stated in Main Text. Details of the protocol are provided in Supplementary Note 8.

## Cyclic S-matrix

An S-matrix of order $n$ can be defined as any matrix within the class of $\{0, 1\}$-valued $n \times n$ matrices that maximizes the determinant[90]. It can be shown that for this S-matrix to exist, $n$ must be either equal to 1, or of the form $4w - 1$ for some positive integer $w$[55]. Moreover, it can be shown that for any S-matrix denoted by $S$[90],

$$S^{-1} = \frac{2}{n+1}\left(2S^T - J\right),\qquad(3)$$

where $J$ denotes the all-ones matrix with size $n \times n$, and $S^T$ denotes the matrix transpose of $S$.

Cyclic S-matrices possess an additional cyclic structure, whereby the initial row determines all subsequent rows via left-wise circular shifts. In particular, letting $S_{i,j}$ denote the element of $S$ with row index $i$ and column index $j$, $S_{i,j}$ satisfies

$$S_{i,j} = S_{0,i+j},\qquad(4)$$

where $\{i, j\} = 0, \ldots, n - 1$, and $i + j$ is interpreted modulo $n$. For imaging, it is desirable that $n$ be factorizable into two parts of approximately equal size.

For the pattern design in SPIRIT, the "twin-prime construction method" was used[37,55]. Given any pair of twin primes $p$ and $q = p + 2$, it is possible to construct a cyclic S-matrix of order $n = pq$. The following functions can then be defined:

$$f(j) = \begin{cases} +1 & \text{if } j \text{ is a quadratic residue (mod } p) \\ 0 & \text{if } j \equiv 0 \,(\text{mod } p) \\ -1 & \text{otherwise} \end{cases} \quad \text{and}\qquad(5)$$

$$g(j) = \begin{cases} +1 & \text{if } j \text{ is a quadratic residue (mod } p) \\ 0 & \text{if } j \equiv 0 \,(\text{mod } p) \\ -1 & \text{otherwise} \end{cases},\qquad(6)$$

from which the elements of $S$ are computed, with $i + j$ interpreted modulo $n$, by

$$S_{i,j} = \begin{cases} 0 & \text{if } [f(i+j) - g(i+j)]g(i+j) = 0 \\ +1 & \text{otherwise} \end{cases}.\qquad(7)$$

## Data interpolation and image reconstruction

The image reconstruction procedure of SPIRIT leverages the fact that the multiplication of a vector by a cyclic S-matrix of appropriate order can be viewed as an operation of 2D discrete convolution[37,56]. In particular, the following equivalence of indexed sums may be observed for expressing the matrix-vector product of an underlying image $x$ with a cyclic S-matrix $S$:

$$y_u = \sum_{v=0}^{n-1} S_{u,v} x_v = \sum_{k=0}^{p-1}\sum_{l=0}^{q-1} S_0(i+k, j+l)x(k,l) = y(i,j).\qquad(8)$$

Here, the notations $y(i,j)$, $x(k,l)$, and $S_0(i+k, j+l)$ are used as respective shorthands for $y_u$, $x_v$, and $S_{0,u+v}$ (i.e., showing the conversion between 2D and 1D indexing schemes via row-major ordering), where $u = qi + j$, $v = qk + l$, and all resulting indices are interpreted as modulo $n$. The salient feature of Eq. (8) is that, since both horizontal and vertical index shifts of $\pm 1$ result in a high degree of similar spatial overlap between encoding patterns and the underlying image, bucket signals collected from the SPIRIT system exhibit 2D smoothness when organized according to row-major ordering. In terms of linear indexing, this feature implies that for any individual bucket signal $y_u$, a high degree of correlation should be expected between values such as of $y_{u\pm1}$ and $y_{u\pm q}$ that neighbor $y_u$ after 2D reshaping.

This expected 2D smoothness gives rise to redundancy that is exploited to achieve a reduction in the size of physical encoding masks used in SPIRIT[36,55]. In particular, the physical masks are designed in such a way that a continuous scan acquires the elements of $y$ in a diagonal fashion within a 2D array, i.e., $y(0,0)$, $y(1,1)$, ..., and $y(m-1, m-1)$, in a compressed data collection (i.e., with $m < n$). This design also allows the measured elements of $y$ to be distributed in an approximately uniform manner (Supplementary Fig. 5e), well-suited for the estimation of non-measured elements via 2D interpolation. More details about the design procedure used to select the desired matrix order $n$ and scan size $m$ are provided in Supplementary Note 5.

Discrete Laplace interpolation[91] was used to estimate the non-measured data points in the 2D-shaped bucket signals by

$$y_u = \frac{1}{4}\left[y_{u+1} + y_{u-1} + y_{u+q} + y_{u-q}\right].\qquad(9)$$

This approach allows a complete $n$-element interpolated data vector $\tilde{y}$ to be constructed as a combination of all bucket signal values both measured and estimated. From the full interpolated data vector $\tilde{y}$, a final recovered image $\tilde{x}$ is obtained via inversion of the full

measurement matrix. Finally, the vector $\tilde{x}$ may be reshaped according to both row-major ordering, as well as separation and rotation to produce images concurring with the two regions in the FOV defined by the windows on the window plate.

**Reporting summary**

Further information on research design is available in the Nature Portfolio Reporting Summary linked to this article.

## Data availability

All data needed to evaluate the findings of this study are present in the paper and Supplementary Information. Raw data are provided with the Source Data Package (https://doi.org/10.5281/zenodo.16790723).

## Code availability

A MATLAB software package is available as Supplementary Software Package (https://doi.org/10.5281/zenodo.16790730).

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

## Acknowledgements

The authors thank Prof. My Ali El Khakani and Prof. Amar Mitiche for fruitful discussions and experimental assistance. The authors also thank all the volunteers for their participation and time. This work was supported in part by the Natural Sciences and Engineering Research Council of Canada (RGPIN-2024-05551, ALLRP 592389-23), the Canada Research Chairs Program (CRC-2022-00119), the INRS Chair in Nano-biophotonics, and the Fonds de Recherche du Québec–Nature et Technologies (203345–Centre d'Optique, Photonique, et Lasers) to J.L.

## Author contributions

C.J. built the experimental setup, performed the experiments, and analyzed the data. P.K. developed the first generation of the reconstruction algorithm. Y.L. assisted in some experiments. J.L. and T.O. initiated the project. J.L. proposed the concept, contributed to experimental design, and supervised the project. C.J. and P.K. drafted the manuscript. All authors revised the manuscript.

## Competing interests

J.L., C.J. and P.K. declare a competing interest related to a US Provisional Patent Application (No. 63/824,609). Y.L. and T.O. declare no competing interests.
