## [Transparent Peer Review file · Nature Communications]

Single-pixel infrared imaging thermometry maps human inner canthi temperature

Corresponding Author: Professor Jinyang Liang

Version 0:

Reviewer comments:

Reviewer #1

(Remarks to the Author)

Major point

The authors have presented an interesting and innovative technology to map the regional temperature close to human inner canthus/canthy. Unfortunately, its merit for physiological applications presented in the manuscript seems to be premature. The manuscript does not show any statistical analysis for the physiological differences claimed by the authors. These differences include the diurnal change of regional temperature (lines 282-292; N=4, repeated measures), temperature related to wearing glasses (lines 296-304; two groups of 4 subjects), and temperature between the genders (lines 270-272; 7 for each gender). Whether the differences (using rather small sample sizes) are statistically significant is uncertain. Without this critical step to validate a real difference, further explanation and discussion of such differences (for examples, lines 304-313 as well as a significant conclusion in the Discussion) using the new technology can be misleading. In the literature, regional temperature differences, for example the diurnal change of temperature, can be observed using infrared camera with a reasonable sample size and validated by a statistical analysis.

Other points

Definition of "temperature of inner canthus/canthy" in the current study needs to be clarified. Inner canthus is defined originally, probably in Latin, as the angle formed by the inner/nasal junction of the upper and lower eyelids. Infrared thermography of the face may regard the temperature of inner canthus as the average (or the maximal) temperature from a circular region close to the junction due to the difficulty in pinpointing the location. Therefore, the region contains the conjunctiva and the eyelids. It seems that the author had excluded the skin (upper eyelid and lower eyelid) for their definition of temperature of inner canthus (lines 32-33 and 259-260). Are the temperatures shown in the figures 3-5 as the averages of the entire rectangles (including the skin) in the figures? As shown, the boundary between the conjunctiva and eyelids can be distinguished using the new technology. How to calculate the temperature from only the conjunctiva? It should be emphasized that thermal regulations of conjunctiva (inside the inner canthus) and eyelids (skin) are different. Having had a clear definition of inner canthus temperature used by the authors, the readers will be able to understand the manuscript better.

While the authors have presented advantages of this new technology over the regular camera-based technology, its limitation is not well discussed. For example, the temperature resolution for this new technology is 0.3°C, which is inferior to the temperature resolution of 0.1°C achievable in commonly used infrared cameras. Another limitation, data acquisition needs approximately 15 seconds. Since a healthy person blinks every 4-5 seconds, fluid dynamics for the thin tear film over the regional ocular surface (inner canthus) can be quite different from an eye intentionally open for 15 seconds. Addressing these issues from various dimensions (e.g., engineering, physiology, and clinical practice) should be helpful to the readers.

(Remarks on code availability)

Reviewer #2

(Remarks to the Author)

This manuscript discusses a method to obtain quantitative thermometry maps of human inner canthi by structured sampling and single-pixel detection. The technique consists on projecting an image of the inner canthi of the human eyes onto a sequence of mask patterns. The light transmitted by the masks is integrated by a long-wave infrared radiation (LWIR) onto

an HgCdTe photodiode. LWIR images of the inner canthi are obtained by an inverse computational imaging technique. The temperature map is obtained from those images after a proper calibration technique.

The paper is relevant, innovative, and very well written. It is relevant because precise human thermometry may have strong implications in health problem solving strategies. Besides, it has many other potential applications, such as security or quality control in industry. It is innovative because technological problems related with LWIR are solved in an ingenious way with a single-pixel detector, avoiding complex pixelated devices. Also, it combines new sampling strategies for single-pixel imaging techniques that solve problems of practical application. In particular, it introduces a clever way to codify 2D-masks from a cyclic S-matrix that allows proper sampling by simply shifting a mask. It shows clearly, perhaps for the first time, how computational imaging techniques based on single-pixel detection can outperform conventional imaging techniques, at least for LWIR. It is well written because it is concise, describes properly previous approaches for thermometry applications, and gives clear and detailed description of the theory involved. Experiments are well designed and results are sound and conclusive.

I would like to comment some thoughts that may be discussed or taken into account:

- 1) The authors discuss several key advantages of using single-pixel detection over 2D pixelated devices for thermography with LWIR. Among them, COMS devices find challenges in practicality and measurement accuracy, as 2D LWIR sensors require cooling systems with efficient heat dissipation. This makes these devices complex and expensive. Authors mention also the 2D LWIR systems are designed to image the entire human face, but the small size and spatial separation of the inner canthi results in a considerably low portion of allocated pixels for temperature mapping. I don't agree with this statement. Optical systems can be designed to magnify a small portion of the face, such as the inner canthi onto the hole sensor. Besides, many CMOS cameras allow to configure multiple ROIs, which will allow to increase versatility and resolution.
- 2) The scanning system is based on two window plates with open areas with the same separation same as that of the two encoded stripes. How can the system be adapted to different geometries of the human face?
- 3) The size of the IR light detector is very small. Just 1mm x 1 mm. Therefore, it should be very difficult to collect efficiently all light passing through the windows. How is this problem solved? This is not clear from the description of the optical system.
- 4) It is written that the spatial resolution is determined by the finite pixel size. That all four orientations of strips in the experimental evaluation exhibit the same spatial resolution despite the anisotropy. But the spatial resolution depends on the magnification of the optical system projecting the object onto the mask patterns. A detailed description of magnification could be useful to the reader.
- 5) Which is the spectral response of the whole optical system?
- 6) Is the measurement error taken into account in Eq. (2)?
- 7) How are the inner canthi identified to measure the temperature? Are they detected automatically from the whole image?

(Remarks on code availability)

Reviewer #3

(Remarks to the Author)

In this work, Jiang et al. report single-pixel infrared imaging thermometry (SPIRIT). SPIRIT senses thermal signals by measuring far-infrared radiation in 2D using a point detector. A new coding strategy is used to aggregate patterns to support diagonal linear scan for compressed sensing and the following image reconstruction without an iterative algorithm. The system enabled spatially resolved detection of temperature for the mapping of human inner canthi with 11x13 pixels and a resolution of 0.3 degree C. The authors used this system to monitor the human temperature cycle throughout the day and the subtle temperature differences induced by glass wearing.

This interesting work separates itself from the literature on single pixel imaging (SPI). Thermal imaging (from the far infrared) is rarely seen in SPI. The imaging techniques in this spectral band are limited by optics and (expensive) detectors, so SPI becomes an attractive technique to circumvent the strict imaging relation and the high cost. This technique offers better performance in temperature resolution than existing thermography. Also, SPIRIT finds an interesting application that suits the technical parameters of SPI. Oftentimes, SPI is criticized for not having the same level of pixel count as a focal array (for instance, CCD/CMOS sensors). But in the case of temperature mapping of inner canthi, the field of view is composed of two separated and small areas. The required number of pixels well matches the inner canthi areas. The work also showcases the capability of SPI in a flexible selection of the field of view and the efficiency of not wasting any pixels (compared to a focal array). I think this work has the value of reference for the community of SPI as well as a broader readership. The work contains both technical development and application, so it is done rather completely. The manuscript is well written. Overall, this is a high quality manuscript. I recommend publishing it in Nature Communications.

That said, I think the authors should still revise it to further clarify the coding strategy. Currently, some places in this part are not easy to follow. Since this is the main conceptual novelty of this work, it deserves to be better presented for general readers who are interested in this work.

- Main Text, Lines 167-170 say that the new coding narrows the gap between unsampled signals (Fig. 2c). Further information can be found in Supp Fig. 5e. However, it is unclear how the new coding strategy would compare to the conventional linear scan. Supplementary Figure 6 only compares the proposed coding strategy with the full sampling case. The authors should expand Supplementary Fig. 6 to compare the reconstructed image quality between the new coding and the conventional linear scan.

- Supplementary Fig. 5f is confusing. I understand that the color boundaries are used to assist the understanding of the flipping motion. However, it can be easily mixed with the colors in Supplementary Fig. 5c and d. Thus, I suggest the authors make this process an animation and add it to Supp Video 1.

- Pattern design should be included in the provided software. It is unclear how "m" is determined using Eqs. (S1)–(S3). I cannot reproduce Supplementary Table 1. A relevant question to my comment above is how the value of "m" changes the reconstruction quality. In Supplementary Table 1, m could also be 52 for 11x13 patterns for a compression ratio of 36.4%. This number is higher than many compression ratios used in compressed sensing-based SPI. Would this value lead to good reconstruction?

(Remarks on code availability)

Version 1:

Reviewer comments:

Reviewer #2

(Remarks to the Author)

The authors have carefully addressed the concerns raised during the review process and have significantly improved the quality and clarity of their manuscript. In particular, they have performed new experiments with an expanded data set, clarified definitions and descriptions, explained experimental details, and provided excellent new videos.

In summary, the authors have demonstrated an exceptional commitment to improving their work, resulting in a manuscript of significantly improved quality. The revisions have addressed all outstanding concerns and raised the manuscript to a standard suitable for publication in Nature Communications. I am confident that this is a sound and strong paper that will attract the attention of Nature Communications readers and provide valuable insights. I therefore strongly recommend its acceptance.

(Remarks on code availability)

Reviewer #3

(Remarks to the Author)

All my concerns have been addressed and I recommend the publication of this work.

(Remarks on code availability)

Response to Reviewers' Comments

We sincerely appreciate the reviewers for their thorough reviews of our work and their insightful and constructive comments, which have helped greatly improve the quality of our manuscript. In this round of revision, we have included detailed descriptions and illustrations to explain and compare the technical features of our work. We have also conducted new experiments to directly address the reviewers' concerns.

In Main Text, these amendments are reflected in the following major points:

- We have updated Fig. 4 and Fig. 5 based on the new experiments and the new statistical analyses.
- We have augmented the sections of Results and Discussion to describe the new results, further clarify SPIRIT's technical details, and expand the prospect of this technique.

As for Supplementary Materials, the major changes include the following points:

- We have expanded Supplementary Note 7 and Supplementary Fig. 6 to present the simulation results using the conventional linear scan and compare them to SPIRIT.
- We have expanded Supplementary Table 2 to include the data from the newly recruited 25 volunteers.
- We have expanded Supplementary Video 1 to illustrate the pattern generation process.
- We have added Supplementary Note 9, Supplementary Figs. 8–10, and Supplementary Table 3 to present detailed statistical analyses of the experimental data.
- We have updated the software package to include the script that produces Supplementary Table 1 and generates the encoding patterns.

In the following, we provide point-by-point responses. The changes in the revised manuscript are highlighted in red.

Reviewer #1

[Comment 0]

Major point

The authors have presented an interesting and innovative technology to map the regional temperature close to human inner canthus/canths.

[Response 0]

We thank the reviewer for recognizing our work as interesting and valuable and appreciate the acknowledgment of its innovative aspects.

[Comment 1]

Unfortunately, its merit for physiological applications presented in the manuscript seems to be premature. The manuscript does not show any statistical analysis for the physiological differences claimed by the authors. These differences include the diurnal change of regional temperature (lines 282-292; N=4, repeated measures), temperature related to wearing glasses (lines 296-304; two groups of 4 subjects), and temperature between the genders (lines 270-272; 7 for each gender). Whether the differences (using rather small sample sizes) are statistically significant is uncertain. Without this critical step to validate a real difference, further explanation and discussion of such differences (for examples, lines 304-313 as well as a significant conclusion in the Discussion) using the new technology can be misleading. In the literature, regional temperature differences, for example the diurnal change of temperature, can be observed using infrared camera with a reasonable sample size and validated by a statistical analysis.

[Response 1]

We thank the reviewer for this valuable feedback and acknowledge the concern regarding the lack of in-depth statistical analysis in the initial submission. In response, we have expanded our dataset and conducted comprehensive statistical analyses to validate the physiological differences discussed in the revised manuscript.

We have added more data from 25 newly recruited volunteers to the revised manuscript. We have given our maximum effort to recruit as many volunteers as possible despite the limitations from the granted time for revision (i.e., 3.5 months from November 8th, 2024), the Christmas/New Year break, and severe winter weather in Canada. Details about the recruitment are summarized in Table R1. All relevant measurement information has been updated in the revised Supplementary

Table 2. These efforts have increased the number of datasets in various categories by up to 6.25 times. While the large-scale physiological test is still beyond our reach, we believe these new datasets have strengthened the demonstration of SPIRIT’s feasibility.

Table R1. Summary of new experiments and analyses presented in the revised manuscript.

Experiment	Number of volunteers		T-test result	Revision
	Original manuscript	Revised manuscript		
Demonstration of temperature mapping	14	39	N/A ^[Note]	Lines 254, 269–270, 272-275 and Fig. 4b in Main Text
Diurnal temperature variation	4	29	Significant	- Lines 285–293, 299, and Fig. 5a in Main Text - Supplementary Note 9 - Supplementary Fig. 8 - Supplementary Table 3
Gender-related temperature difference	4	28	Significant	- Lines 301–311 and Fig. 5c in Main Text - Supplementary Note 9 - Supplementary Fig. 9
Glasses-induced temperature difference	8	28	Significant	- Lines 312–315 and Fig. 5e in Main Text - Supplementary Note 9 - Supplementary Fig. 10

^[Note] No statistical analysis was performed for the demonstration of temperature mapping because it aimed to experimentally prove the concept of SPIRIT.

To address the reviewer’s concern about statistical validation, we have conducted rigorous tests for statistical significance with updates of Fig. 5 in Main Text, as well as analyses detailed in Supplementary Note 9 and Supplementary Figs. 8–10. In brief, the SPIRIT measured temperatures were first divided into distinct time windows to mitigate diurnal variations. Then, we compared groups that differed only in one variable (i.e., gender or glasses-wearing). To confirm these differences, we first verified data normality with the Kolmogorov–Smirnov test. Next, we applied a two-sample t-test to establish significant differences ($p < 0.05$) between groups, followed by a

right-tailed t-test to confirm that one group consistently exhibited higher temperatures. These statistical analyses have concluded that the observed temperature differences from diurnality, gender, and glasses-wearing are statistically significant.

In addition, we appreciate the reviewer's observation regarding the use of infrared cameras to detect regional temperature differences, such as diurnal changes, with reasonable sample sizes validated by statistical analysis. We fully acknowledge this established capability of infrared cameras, as noted in the literature (cited as Refs. 24 and 25, in Main Text). However, the primary aim of SPIRIT is not to replicate what infrared cameras can achieve but to provide a more practical and economical solution specifically for temperature mapping of human inner canthi. As noted in Lines 50–55 of Main Text, infrared cameras are hampered by expensive cooling systems that increase system complexity and induce vibrations. They also face the challenges of low pixel allocation for inner-canthus areas, which results in background blending and temperature inaccuracies. In contrast, as described in Discussion (Lines 354–356 in Main Text), SPIRIT dedicates its entire field of view (FOV) to the inner canthi, which enhances measurement efficiency and minimizes background blending. Its passive, non-contact, and cost-effective design could further facilitate large-scale screening and precise public health assessments.

[Comment 2]

Other points

Definition of “temperature of inner canthus/canthi” in the current study needs to be clarified. Inner canthus is defined originally, probably in Latin, as the angle formed by the inner/nasal junction of the upper and lower eyelids. Infrared thermography of the face may regard the temperature of inner canthus as the average (or the maximal) temperature from a circular region close to the junction due to the difficulty in pinpointing the location. Therefore, the region contains the conjunctiva and the eyelids. It seems that the author had excluded the skin (upper eyelid and lower eyelid) for their definition of temperature of inner canthus (lines 32-33 and 259-260). Are the temperatures shown in the figures 3-5 as the averages of the entire rectangles (including the skin) in the figures? As shown, the boundary between the conjunctiva and eyelids can be distinguished using the new technology. How to calculate the temperature from only the conjunctiva? It should be emphasized that thermal regulations of conjunctiva (inside the inner canthus) and eyelids (skin) are different.

Having had a clear definition of inner canthus temperature used by the authors, the readers will be able to understand the manuscript better.

[Response 2]

We appreciate the reviewer for offering these insightful details. We also thank the reviewer for highlighting the importance of clarifying the definition of “temperature of the inner canthus/canths”. In this revised manuscript, we have added a detailed description of the inner canths (Lines 261–263 in Main Text) that uses the definition supplied by the reviewer.

The reviewer also raises an important point about previous studies using infrared thermography, where the temperature of the inner canths is often calculated as the average or maximal value from a circular region near the junction of the eyelids due to the difficulty in pinpointing their precise locations. As the reviewer noted, the thermal regulation of the inner canths differs from that of the surrounding eyelids. Consequently, such an approximate circular mapping can lead to inaccurate thermal readings, introducing uncertainty into the studies of inner canths thermal regulation.

In contrast, SPIRIT addresses this challenge from two aspects. First, a visible camera is used to delineate both inner canths. Second, SPIRIT allocates more pixels in the inner canths areas than infrared cameras, which allows the selection of pixels exclusively from the targeted areas. This capability enables SPIRIT to offer more precise localization of the inner canths and more accurate temperature reporting that reflects the thermal regulation of the inner canths. In this regard, it is important to note that SPIRIT *excludes* the eyelid skin when presenting the temperature of the inner canths. The temperatures shown in Figs. 4 and 5 in Main Text are *not* averaged over the entire FOV (referred to as “rectangles” in the reviewer’s comment). Instead, they are the averaged values specifically calculated for the inner canths areas. To clarify this point, we have added a clarification in Lines 267–268 of Main Text.

[Comment 3]

While the authors have presented advantages of this new technology over the regular camera-based technology, its limitation is not well discussed. For example, the temperature resolution for this new technology is 0.3°C, which is inferior to the temperature resolution of 0.1°C achievable in commonly used infrared cameras.

[Response 3]

We thank the reviewer for acknowledging the advantages of SPIRIT over regular infrared thermography technologies. However, we have a different opinion from the reviewer's statement that SPIRIT's temperature resolution of 0.3°C is inferior to that of commonly used infrared cameras. Based on our interpretation, "commonly used infrared cameras" refer to consumer-grade or entry-level thermal imaging systems typically priced under \$1,000. Such devices are widely deployed in airports, transportation hubs, and other mass screening environments. These cameras typically use uncooled microbolometer sensors, which cannot reach a temperature resolution of 0.1°C. Although they can detect small temperature differences, their absolute accuracy is generally limited to the degree level, as seen in models from FLIR or SEEK Thermal [R1, R2]. As written in the Introduction section (see Lines 50–52 in Main Text), to reach 0.1°C temperature resolution, it is necessary to use cooled infrared cameras, which require complex cooling and thus significantly high cost (typically in the range from 10,000\$ to 50, 000\$) [R3]. In contrast, SPIRIT achieves comparable temperature accuracy without the need for a complex cooling system.

Nonetheless, we agree with the reviewer that the limitations of SPIRIT should be put forward for readers to have a thorough and complete assessment of this technology. In particular, SPIRIT requires precise alignment, image reconstruction, and a relatively long acquisition time. In the revised manuscript, these points have been added to the Discussion section (please see Lines 370–374 in Main Text).

[Comment 4]

Another limitation, data acquisition needs approximately 15 seconds. Since a healthy person blinks every 4-5 seconds, fluid dynamics for the thin tear film over the regional ocular surface (inner canthus) can be quite different from an eye intentionally open for 15 seconds. Addressing these issues from various dimensions (e.g., engineering, physiology, and clinical practice) should be helpful to the readers.

[Response 4]

We thank the reviewer for highlighting the potential influence of fluid dynamics of the tear film on the inner-canthus temperature during data acquisition. We addressed this concern in the newly acquired 25 datasets by using a reference thermal camera to monitor the thermal behavior of the inner canthi throughout the entire acquisition period. As an example, the result of the volunteer

V22 is shown in Fig. R1. The tracked temperature of both inner canthi barely fluctuated, reflected by the small standard deviations of $0.06^{\circ}\text{C}/0.07^{\circ}\text{C}$ (left/right inner canthus), $0.02^{\circ}\text{C}/0.03^{\circ}\text{C}$, $0.03^{\circ}\text{C}/0.04^{\circ}\text{C}$, and $0.03^{\circ}\text{C}/0.04^{\circ}\text{C}$ across the four time windows. Measured data for all 25 newly recruited volunteers are summarized in Table R2. This level of temperature fluctuation suggests that the possible impact of fluid dynamics of the tear film on SPIRIT’s temperature reconstructions is negligible.

That said, we acknowledge that SPIRIT’s relatively long acquisition time could cause discomfort to volunteers. This limitation arises from the use of a motorized mechanical translation stage in the current SPIRIT system. To address this issue, as discussed in Lines 379–380 in Main Text, we plan to implement a high-speed rotating mask to enhance the imaging speed to video rate. In this way, the potential influence of tear film will be eliminated on temperature measurements.

Fig. R1. Temperature tracking of volunteer V22 using the reference thermal camera. a Representative frames captured by the reference thermal camera at four time points in a day. Red boxes indicate SPIRIT’s FOV. **b** Corresponding temperature tracking results of inner canthi through the data acquisition. LIC and RIC: left inner canthus and right inner canthus, respectively.

Table R2. Temperature tracking information on the newly recruited volunteers.
(LIC: left inner canthus; RIC: right inner canthus)

Index of volunteers	Measurement time	LIC fluctuation (°C)	RIC fluctuation (°C)
V15	10:06	0.04	0.05
	12:04	0.04	0.09
	14:05	0.07	0.06
	15:59	0.07	0.06
V16	9:58	0.09	0.05
	11:38	0.03	0.03
	13:46	0.07	0.05
	15:47	0.05	0.05
V17	9:46	0.07	0.03
	11:39	0.05	0.04
	13:31	0.07	0.05
	15:43	0.10	0.07
V18	9:56	0.07	0.07
	11:35	0.07	0.07
	13:43	0.06	0.06
	15:36	0.07	0.06
V19	9:42	0.07	0.05
	11:59	0.03	0.04
	13:48	0.07	0.06
	15:50	0.08	0.06
V20	9:50	0.05	0.04
	12:04	0.06	0.07
	14:02	0.08	0.08
	15:58	0.08	0.07
V21	9:57	0.03	0.06
	11:42	0.03	0.02
	13:36	0.04	0.04
	15:41	0.05	0.05
V22	10:04	0.06	0.07
	11:47	0.02	0.03
	13:59	0.03	0.04
	15:34	0.03	0.04
V23	9:33	0.05	0.04
	11:30	0.05	0.04
	13:31	0.04	0.06
	15:18	0.02	0.02
V24	9:46	0.03	0.03
	11:41	0.11	0.12
	13:40	0.03	0.04
	13:05	0.04	0.07

V25	10:04	0.05	0.05
	11:56	0.08	0.06
	14:00	0.07	0.05
	15:50	0.07	0.09
V26	10:12	0.06	0.06
	12:01	0.08	0.07
	14:07	0.05	0.06
	15:53	0.06	0.04
V27	9:59	0.04	0.04
	11:52	0.17	0.13
	13:42	0.04	0.06
	15:32	0.11	0.12
V28	9:55	0.04	0.05
	11:54	0.07	0.05
	13:35	0.04	0.05
	15:40	0.05	0.07
V29	9:50	0.07	0.06
	11:41	0.10	0.16
	13:52	0.05	0.07
	15:59	0.05	0.08
V30	9:50	0.04	0.05
	12:12	0.05	0.04
	13:54	0.04	0.06
	15:27	0.14	0.16
6V31	9:35	0.04	0.06
	11:33	0.07	0.05
	13:30	0.11	0.09
	15:32	0.07	0.07
V32	9:40	0.14	0.15
	11:32	0.07	0.07
	13:55	0.04	0.04
	15:22	0.03	0.05
V33	9:47	0.13	0.09
	12:11	0.15	0.11
	14:07	0.08	0.15
	15:50	0.12	0.18
V34	10:12	0.17	0.08
	11:29	0.08	0.09
	13:54	0.05	0.09
	15:37	0.05	0.07
V35	10:18	0.08	0.15
	11:30	0.15	0.08
	13:49	0.05	0.05
	15:40	0.10	0.15
V36	10:22	0.02	0.03
	11:32	0.07	0.05

	13:55	0.04	0.05
	15:42	0.08	0.12
V37	10:00	0.05	0.06
	11:57	0.12	0.15
	13:34	0.05	0.08
	15:37	0.07	0.08
V38	9:44	0.06	0.06
	11:37	0.15	0.12
	13:40	0.05	0.08
	15:27	0.03	0.04
V39	9:39	0.05	0.04
	11:40	0.16	0.16
	13:42	0.06	0.07
	15:43	0.03	0.04

Reviewer #2

[Comment 0]

This manuscript discusses a method to obtain quantitative thermometry maps of human inner canthi by structured sampling and single-pixel detection. The technique consists on projecting an image of the inner canthi of the human eyes onto a sequence of mask patterns. The light transmitted by the masks is integrated by a long-wave infrared radiation (LWIR) onto an HgCdTe photodiode. LWIR images of the inner canthi are obtained by an inverse computational imaging technique. The temperature map is obtained from those images after a proper calibration technique.

The paper is relevant, innovative, and very well written. It is relevant because precise human thermometry may have strong implications in health problem solving strategies. Besides, it has many other potential applications, such as security or quality control in industry. It is innovative because technological problems related with LWIR are solved in an ingenious way with a single-pixel detector, avoiding complex pixelated devices. Also, it combines new sampling strategies for single-pixel imaging techniques that solve problems of practical application. In particular, it introduces a clever way to codify 2D-masks from a cyclic S-matrix that allows proper sampling by simply shifting a mask. It shows clearly, perhaps for the first time, how computational imaging techniques based on single-pixel detection can outperform conventional imaging techniques, at least for LWIR. It is well written because it is concise, describes properly previous approaches for thermometry applications, and gives clear and detailed description of the theory involved. Experiments are well designed and results are sound and conclusive.

[Response 0]

We thank the reviewer for recognizing the innovation and impact of our work in human thermometry and other potential applications, as well as for highlighting the clarity of the manuscript and the sound design of our experiments.

[Comment 1]

I would like to comment some thoughts that may be discussed or taken into account:

1) The authors discuss several key advantages of using single-pixel detection over 2D pixelated devices for thermography with LWIR. Among them, COMS devices find challenges in practicality and measurement accuracy, as 2D LWIR sensors require cooling systems with efficient heat dissipation. This makes these devices complex and expensive. Authors mention also the 2D LWIR

systems are designed to image the entire human face, but the small size and spatial separation of the inner canthi results in a considerably low portion of allocated pixels for temperature mapping. I don't agree with this statement. Optical systems can be designed to magnify a small portion of the face, such as the inner canthi onto the hole sensor. Besides, many CMOS cameras allow to configure multiple ROIs, which will allow to increase versatility and resolution.

[Response 1]

We thank the reviewer for the valuable comments regarding the use of optical magnification and multiple regions-of-interest (ROIs) configurations for 2D LWIR sensors. We provide our clarifications below:

First, we address the reviewer's comment regarding the ability to magnify the inner canthi onto the "hole" sensor (We note that the use of "hole" here might be a typographical error for "whole"). While magnification can indeed fill more of the sensor area with the target, the fundamental challenge remains: since the two inner canthi are spatially separated, a single snapshot from a 2D sensor inherently captures the area between them (i.e., intercanthal area). Thus, even with optimal magnification, the pixels corresponding to the intercanthal area are essentially unused for temperature measurement. As mentioned in Lines 121–122 in Main Text, the typical intercanthal distance is approximately 30 mm, and each inner canthus itself measures about 3–5 mm in length [R4]. These facts further suggest a large number of unused pixels for the intercanthal area. In other words, the spatial separation of the two inner canthi leads to an inherently and unavoidably low efficiency in the use of the sensor area.

Second, we appreciate the reviewer's mention that many CMOS cameras allow the configuration of multiple ROIs. This capability indeed enables the camera to transmit data exclusively from user-specified pixels on the sensor, which could lead to a versatile selection of interested spatial features with higher data rates while accommodating the readout bandwidth limit of the camera [R5]. However, the function of multi-ROIs does not affect the spatial resolution imposed by the diffraction limit of the deployed optics. If the reviewer defined "resolution" as the image size, then ROIs would reduce the resolution because data from non-selected ROIs are disregarded. Consequently, multiple ROIs would not increase resolution. Most importantly, multi-ROIs do not bypass the inherent limitation of having only a small portion of the sensor's pixels devoted to each inner canthus.

In summary, we acknowledge that the reviewer’s suggestions—higher magnification and multi-ROIs—can help improve the use of pixels on a 2D sensor for the inner canthi to some extent. Nonetheless, these approaches do not change the fact that the 2D sensor inevitably allocates pixels to the irrelevant area between the two inner canthi. On the contrary, by dedicating all the pixels to the two inner canthi, SPIRIT outperforms traditional 2D sensors from the perspective of resource allocation efficiency.

[Comment 2]

2) The scanning system is based on two window plates with open areas with the same separation same as that of the two encoded stripes. How can the system be adapted to different geometries of the human face?

[Response 2]

We appreciate the reviewer’s question regarding the adaptation of our scanning system to different facial geometries. First, we would like to clarify that SPIRIT’s setup includes a *single* window plate with *two* open rectangular areas (please see the photo in Supplementary Fig. 1b). The reviewer’s point of “two window plates” appears to be a misunderstanding. The two open windows on this plate align with the encoded stripes on the mask plate (please see the photo in Supplementary Fig. 1a).

SPIRIT can accommodate most geometries of the human face for temperature mapping of the inner canthi. As noted in Main Text (Lines 111–112 and 120–121), the two open windows are positioned with a -45° and 45° of rotation, which extends the lateral coverage. Combined with visible camera-assisted FOV co-registration, this arrangement can tolerate intercanthal distances between 30 mm and 54.75 mm, which covers approximately 100% of the population [R6]. Other potential variations—such as facial curvature—have a limited impact on SPIRIT’s performance.

In extreme cases where the intercanthal distance exceeds the defined range, the subjects can still have their temperature measured by performing two separate scans, albeit with some inconvenience. Furthermore, a zoom ZnSe lens could be designed to adjust the magnification ratio according to the subject’s facial geometry so that the effective intercanthal distance imaged onto the mask plate remains constant (i.e., 7.5 mm), which would further enhance SPIRIT’s adaptability to diverse facial geometries. In the revised manuscript, this point has been incorporated in Lines 381–384 in Main Text.

[Comment 3]

3) The size of the IR light detector is very small. Just 1 mm x 1 mm. Therefore, it should be very difficult to collect efficiently all light passing through the windows. How is this problem solved? This is not clear from the description of the optical system.

[Response 3]

SPIRIT's data acquisition does not aim to form a spatially resolved image. Rather, as described in Lines 115–118 in Main Text, for each encoding mask, *all* the transmitted thermal radiation is collected by the second ZnSe lens (see Fig. 1 in Main Text) to generate one bucket signal. Hence, the small size of the deployed photodiode does not pose any technical difficulty in data acquisition.

[Comment 4]

4) It is written that the spatial resolution is determined by the finite pixel size. That all four orientations of strips in the experimental evaluation exhibit the same spatial resolution despite the anisotropy. But the spatial resolution depends on the magnification of the optical system projecting the object onto the mask patterns. A detailed description of magnification could be useful to the reader.

[Response 4]

The reviewer is correct that spatial resolution depends on the magnification of the optical system that images the object onto the encoding masks. In a conventional imaging system, the spatial resolution is ultimately determined by the system's numerical aperture (NA) and the operating wavelength. Nonetheless, when the point spread function (PSF) is smaller than the encoding pixel size, individual PSFs cannot be resolved while the individual encoding pixel becomes the smallest resolution unit, hence defining the spatial resolution.

In the SPIRIT system, the first ZnSe lens has an NA of 0.25, and the operating wavelength is $\sim 10\ \mu\text{m}$. The diffraction-limited resolution is estimated to be $24.4\ \mu\text{m}$, much smaller than the encoding pixel size of $250\ \mu\text{m}$. Hence, the spatial resolution at the intermediate image plane is determined by the encoding pixel size. By considering the 4:1 magnification ratio as well as the anisotropy due to the finite pixel size (please see Lines 105–107 and 216–217 in Main Text), at the object plane, SPIRIT's spatial resolution is $\sim 1\ \text{mm}$, which is consistent with the experimental results (please see Figs. 3c–f in Main Text).

[Comment 5]

5) Which is the spectral response of the whole optical system?

[Response 5]

We thank the reviewer for the inquiry of this useful parameter. Table R3 compiles the key spectral properties of the SPIRIT system's components.

Table R3. Spectral characteristics of the components in the SPIRIT system.

Component	Part number	Transmission spectrum
First ZnSe lens	Edmund Optics, #39-532	0.6 μm –18 μm
Second ZnSe lens	Yoseen Infrared, #307B/1.0	8 μm –12 μm
Photodiode	Thorlabs, PDAVJ10	2 μm –10.6 μm

Limited by the transmission spectra of the second ZnSe lens and the photodiode, SPIRIT's responsive spectrum is constrained to the 8–10.6 μm LWIR band. This range aligns well with the emissivity peak of the human body [R7], which makes SPIRIT well suited for human thermometry and related thermal imaging applications. This information has been added to the revised manuscript (please see Lines 118–119 in Main Text).

[Comment 6]

6) Is the measurement error taken into account in Eq. (2)?

[Response 6]

Yes. As shown in Eq. (1) in Main Text, we incorporate the measurement error Δ directly into measured bucket signals \tilde{x} . In Eq. (2), since the measured temperature map T_m is derived from \tilde{x} , the measurement error propagates to the final results. Thus, Δ has been taken into account in Eq. (2).

[Comment 7]

7) How are the inner canthi identified to measure the temperature? Are they detected automatically from the whole image?

[Response 7]

As described in Lines 260–263 in Main Text, SPIRIT co-registers both inner canthi imaged by single-pixel imaging and by a visible camera. Thus, each inner-canthus area can be delineated on the temperature map. *Only* the pixels in the inner-canthus area were used for the ensuing analysis, hence ensuring high accuracy in temperature sensing. In the current SPIRIT system, the inner canthi are not detected automatically from the thermal image alone. Fully automated detection represents an interesting future research direction and has been added in the revised manuscript (please see Lines 377–379 in Main Text).

Reviewer #3

[Comment 0]

In this work, Jiang et al. report single-pixel infrared imaging thermometry (SPIRIT). SPIRIT senses thermal signals by measuring far-infrared radiation in 2D using a point detector. A new coding strategy is used to aggregate patterns to support diagonal linear scan for compressed sensing and the following image reconstruction without an iterative algorithm. The system enabled spatially resolved detection of temperature for the mapping of human inner canthi with 11x13 pixels and a resolution of 0.3 degree C. The authors used this system to monitor the human temperature cycle throughout the day and the subtle temperature differences induced by glass wearing.

This interesting work separates itself from the literature on single pixel imaging (SPI). Thermal imaging (from the far infrared) is rarely seen in SPI. The imaging techniques in this spectral band are limited by optics and (expensive) detectors, so SPI becomes an attractive technique to circumvent the strict imaging relation and the high cost. This technique offers better performance in temperature resolution than existing thermography. Also, SPIRIT finds an interesting application that suits the technical parameters of SPI. Oftentimes, SPI is criticized for not having the same level of pixel count as a focal array (for instance, CCD/CMOS sensors). But in the case of temperature mapping of inner canthi, the field of view is composed of two separated and small areas. The required number of pixels well matches the inner canthi areas. The work also showcases the capability of SPI in a flexible selection of the field of view and the efficiency of not wasting any pixels (compared to a focal array). I think this work has the value of reference for the community of SPI as well as a broader readership. The work contains both technical development and application, so it is done rather completely. The manuscript is well written. Overall, this is a high quality manuscript. I recommend publishing it in Nature Communications.

That said, I think the authors should still revise it to further clarify the coding strategy. Currently, some places in this part are not easy to follow. Since this is the main conceptual novelty of this work, it deserves to be better presented for general readers who are interested in this work.

[Response 0]

We appreciate the reviewer for their thoughtful feedback and for recognizing our work as a valuable contribution to SPI as well as a broader readership with relevance to both technical development and applications. We also thank the reviewer's suggestion to clarify the coding

strategy and have revised the manuscript to enhance its accessibility, with details addressed in our response to Comment 3.

[Comment 1]

Main Text, Lines 167-170 say that the new coding narrows the gap between unsampled signals (Fig. 2c). Further information can be found in Supp Fig. 5e. However, it is unclear how the new coding strategy would compare to the conventional linear scan. Supplementary Figure 6 only compares the proposed coding strategy with the full sampling case. The authors should expand Supplementary Fig. 6 to compare the reconstructed image quality between the new coding and the conventional linear scan.

[Response 1]

As shown in Supplementary Fig. 5e and Supplementary Video 1, the conventional linear scan is defined using the same sampling ratio as SPIRIT but simply collects the bucket signals sequentially from 0 to 90. Consequently, the unsampled data points in the 2D-shaped bucket signals lack sufficient neighboring samples for effective interpolation employed in SPIRIT (detailed in Fig. 2c, Supplementary Note 6, and Supplementary Fig. 5e).

To demonstrate this point, we have simulated the image reconstruction using encoding masks of the conventional linear scan. The results have been added to Supplementary Figs. 6a, c, and d. The results show that the conventional linear scan fails to reconstruct the image of the objects. This description has been added in Supplementary Note 7.

[Comment 2]

Supplementary Fig. 5f is confusing. I understand that the color boundaries are used to assist the understanding of the flipping motion. However, it can be easily mixed with the colors in Supplementary Fig. 5c and d. Thus, I suggest the authors make this process an animation and add it to Supp Video 1.

[Response 2]

We thank the reviewer for the suggestion and have adopted it. In particular, we have created a new animation that illustrates both the flipping motion and the aggregation process of the encoding patterns to construct the encoding stripe. This animation has been added to Supplementary Video

1. The different colors used for the boxes in Supplementary Fig. 5f have also been replaced by just the red color.

[Comment 3]

Pattern design should be included in the provided software. It is unclear how “ m ” is determined using Eqs. (S1)–(S3). I cannot reproduce Supplementary Table 1. A relevant question to my comment above is how the value of “ m ” changes the reconstruction quality. In Supplementary Table 1, m could also be 52 for 11x13 patterns for a compression ratio of 36.4%. This number is higher than many compression ratios used in compressed sensing-based SPI. Would this value lead to good reconstruction?

[Response 3]

We thank the reviewer for their constructive criticism, and in particular for their efforts in seeking to reproduce our data in Supplementary Table 1. We address these comments in four parts, concerning (1) the inclusion of additional pattern design codes, (2) the reproduction of our data in Supplementary Table 1, (3) the effect of sampling number m on reconstruction quality, and finally (4) the relatively high compression ratio used in our chosen pattern design.

First, to address the inclusion of pattern design codes, our supplementary software package now includes a MATLAB script (`generate_SPIRIT_plates.m`) that outputs an interactive graphical figure containing the design information for the mask and window plates used in SPIRIT. This interactive figure, when produced using a running instance of the MATLAB environment, is suitable for export and inspection in a variety of image file formats. A screenshot of the interactive figure produced by this script is shown in Fig. R2.

Fig. R2. Screenshot of an interactive MATLAB figure of the designs of SPIRIT’s mask plate and window plate.

Second, to address the reproduction of the data of Supplementary Table 1, we have now provided additional details to our mathematical discussion contained in Supplementary Note 5. Moreover, we have now included the computer codes used for the construction of the data in Supplementary Table 1 in our software package (`generate_supp_table_1.m`). The operation of our code is equivalent to the mathematical description we have provided encompassing Eqs. (S1)–(S3). To illustrate this consistency (adopting the notation of Supplementary Note 5 for brevity), we here give an example for the case $a = 11$, $b = 13$ (so that $n = ab = 143$), and $m = 91$ that was used to generate the mask and window plates used by SPIRIT. For these parameters, justification of the choice of $m = 91$ arises from the observation that for $r = 7$, we have $d_r = \lceil a/r \rceil$. We demonstrate this calculation as follows. We first generate the set of values stipulated by Eq. (S2), consisting of the values produced by the function $\text{mod}(14t, 11)$ for inputs $0 \leq t < r = 7$. In order, these values are: 0, 3, 6, 9, 1, 4, 7. Arranging these values in strictly ascending order then produces the sequence $\tilde{a}_0, \dots, \tilde{a}_6$: 0, 1, 3, 4, 6, 7, 9, from which the application of Eq. (S3) yields $d_7 = 2$. Since $\lceil a/r \rceil = \lceil 11/7 \rceil = 2$ also for the ideal size of the largest cyclic gap, our design-search algorithm thus highlights the value of $r = 7$, corresponding to a sampling number $m = rb = 7 \times 13 = 91$.

By searching over all possible values of $18 \leq n \leq 300$ amenable to cyclic S-matrix construction, as well as factorizations $n = ab$ giving rise to rectangular imaging sizes with aspect ratios satisfying $1/2 < a/b < 2$, the data reported in Supplementary Table 1 summarizes particular values of $m = \tilde{r}b$ that corresponded to values of \tilde{r} satisfying $d_{\tilde{r}} = \lceil a/\tilde{r} \rceil$, with the latter equation verified according to calculations similar as to what we have shown above. However, in selecting values of m for inclusion in Supplementary Table 1, an additional rule was applied: in cases where multiple values of r produced equivalent values of d_r , only the smallest values of r were retained (the details of this additional rule have now been included in Supplementary Note 5). This evaluation procedure for diagonally scanned designs is illustrated in Fig. R3 for the two imaging sizes $(a, b) = (11, 13)$ and $(a, b) = (13, 11)$, directly reflecting the corresponding data reported in Supplementary Table 1. As Fig. R3 shows, our design-search criteria work to identify the 2D sampling distributions that minimize m for a given level of uniformity quantified by the first column's largest cyclic gap.

Fig. R3. Examples of design search evaluations of diagonal scanning designs. **a** Plot of the largest cyclic gap size as a function of fill level parameter r encountered by the diagonal scanning design with imaging dimensions $(a, b) = (11, 13)$ used for SPIRIT (marked by the blue solid line), as compared to the ideal lower bound $\lceil a/r \rceil$ (marked by the grey dashed line). Highlighted locations (marked by the red dots) show values of r corresponding to optimal sampling distributions for each value of d_r , with corresponding insets used to visualize the distribution of sampled (white) and non-sampled (black) bucket signals used as priors for 2D interpolation. **b** As in **(a)** but for a diagonal scan design with exchanged imaging dimensions $(a, b) = (13, 11)$, exhibiting a fill pattern that is sub-optimal for most under-sampling scenarios.

Third, concerning the effect of changes to the sampling number m on reconstruction quality, we offer the following comments. For the reconstruction algorithm used by SPIRIT, the key form of data redundancy exploited to achieve compressive sensing is the 2D smoothness exhibited by natural images when interrogated by the encoding patterns of cyclic S-matrices. Since the operation of 2D interpolation is to furnish estimations of unknown data points that are embedded in neighborhoods of known data points, increases in the sampling number m for the discrete and bounded 2D interpolation problem faced by SPIRIT image reconstruction simultaneously serve to decrease the amount of unknown information to be obtained via estimation, while increasing the amount of known information that contributes to reconstruction. These two effects, combined with the highly uniform embedding of known and unknown data furnished by diagonal scanning with correctly designed encoding patterns, provide compelling grounds for the expectation that reconstruction quality should strictly increase with increasing sampling number for SPIRIT, similar to the behavior shown by other compressive approaches to single-pixel imaging.

Although we believe that these observations successfully explain the main effect of sampling number m on reconstruction quality for SPIRIT, we do acknowledge that the trade-off between acquisition time and thermometric accuracy involved in SPIRIT's design would benefit from further quantitative study. Nonetheless, the complexities and additional criteria required for such evaluations, we believe, place this investigation beyond the scope of the present work.

Finally, we address the comparatively high compression ratio used in our chosen pattern design. First, the reviewer is correct in pointing out that, according to our design search summary in Supplementary Table 1, a sampling number of $m = 52$ (corresponding to 36.4% compression) would be consistent with a uniform distribution of sampled values for the 2D interpolation used by SPIRIT's reconstruction algorithm. We acknowledge that the full performance trade-off of the used encoding patterns with respect to the sampling ratio remains unclear at this time. Although such a reduction in compression ratio (and thus linear scan extent) may ultimately be compatible with the requirements of human core temperature measurement envisioned for SPIRIT, we believe that the prioritization of the full space utilization of our mask and window plates is a reasonable design rule, naturally allowing reductions of sampling ratio to be traded off with increases to spatial resolution. However, given the relatively low spatial resolution requirements demanded for inner canthi temperature measurement, we believe that favoring the maximization of sampling ratio

(restricted by linear scan extent) for a fixed imaging resolution constitutes the primary design principle for our SPIRIT architecture.

Regardless, the reviewer is also correct in observing that, even for the $m = 52$ design, the compression ratios used by SPIRIT are large relative to those achieved by other SPI systems utilizing compressive-sensing-based reconstruction (for instance, we are aware of the 3% sampling ratios for 256×256 video reported by the Fourier domain regularized inversion method reported in [R8]). Firstly, although SPIRIT's reconstruction algorithm fully allows for the attempted recovery of $\leq 3\%$ sampled data, by targeting higher sampling ratios, the accuracy of SPIRIT data is enhanced, thus supporting SPIRIT's key application to human core temperature measurement. Second, in contrast to the majority of SPI systems which are based on the use of commercial digital micromirror devices (DMDs), the application of SPIRIT to long-wave infrared measurement imposes the requirement of encoding via transmissive masks, thus precluding approaches to highly compressive SPI that utilize basis scanning such as [R8]. Under this fundamental constraint, we believe that our use of aggregated cyclic S-matrix patterns deployed via diagonal scanning constitutes a competitive, well-considered, and mathematically optimized design for compressive SPI in the far infrared based on linear mechanical scanning. Consequently, we believe that the comparatively high sampling ratios used in SPIRIT, rather than signaling a sub-optimal design, in fact, represent a natural design outcome stemming from the challenging imaging problem that SPIRIT aims to solve.

References in Response

- [R1] FLIR Systems. *Professional Thermal Camera: FLIR T540, specifications*, <https://www.flir.com/products/t540/?segment=solutions&srsltid=AfmBOoqVJgNL4YQnboi2aLt0GLZndy1yG0X_NRtVUQNuGdPnimA2ZTN7&vertical=condition+monitoring> Accessed on: 2025/02/11.
- [R2] SEEK Thermal. *Seek Thermal InspectionCAM Thermal Camera*, <<https://www.pass-thermal.co.uk/seek-thermal-inspectioncam-thermal-camera>> Accessed on: 2025/02/11.
- [R3] Gao, Y., Zhang, B., Chen, L., Xu, B. & Gu, G. Thermal design and analysis of the high resolution MWIR/LWIR aerial camera. *Optik* **179**, 37-46 (2019).
- [R4] Shetty, S. K., Malli, P., D'Souza, J., Shenoy, K., Chunduri, S. T., & Fernandes, K. Inner canthal distance, inter pupillary width, and golden proportion, as predictors of width of the maxillary central incisors--an in vivo study. *Journal of Evolution of Medical and Dental Sciences* **10**, 1650-1656(2021).
- [R5] Cheng, B., Cui, L., Jia, W., Zhao, W. & Gerhard, P. H. Multiple region of interest coverage in camera sensor networks for tele-intensive care units. *IEEE Transactions on Industrial Informatics* **12**, 2331-2341 (2016).
- [R6] Farkas, L. G. *et al.* International anthropometric study of facial morphology in various ethnic groups/races. *Journal of Craniofacial Surgery* **16**, 615-646 (2005).
- [R7] Ring, E. & Ammer, K. Infrared thermal imaging in medicine. *Physiological Measurement* **33**, R33 (2012).
- [R8] Czajkowski, K. M., Pastuszczyk, A. & Kotyński, R. Real-time single-pixel video imaging with Fourier domain regularization. *Optics Express* **26**, 20009-20022 (2018).

Response to Reviewers' Comments

Reviewer #2

[Comment]

The authors have carefully addressed the concerns raised during the review process and have significantly improved the quality and clarity of their manuscript. In particular, they have performed new experiments with an expanded data set, clarified definitions and descriptions, explained experimental details, and provided excellent new videos.

In summary, the authors have demonstrated an exceptional commitment to improving their work, resulting in a manuscript of significantly improved quality. The revisions have addressed all outstanding concerns and raised the manuscript to a standard suitable for publication in Nature Communications. I am confident that this is a sound and strong paper that will attract the attention of Nature Communications readers and provide valuable insights. I therefore strongly recommend its acceptance.

[Response]

We thank the reviewer for the positive assessment and strong recommendation. We appreciate the recognition of our efforts to expand the dataset, perform new experiments, clarify definitions and descriptions, add experimental details, and provide a new video. We are pleased that the revision addresses all concerns.

Reviewer #3

[Comment]

All my concerns have been addressed and I recommend the publication of this work.

[Response]

We thank the reviewer for acknowledging that we have addressed all concerns and for supporting the publication of this manuscript.